# Assessment of Modified Culture Conditions for Fungal Bio-Oxidation of Sulfidic Gold Ores Performed at Circumneutral pH

Guillermo Hein, Harshit Mahandra * and Ahmad Ghahreman

Hydrometallurgy and Environment Laboratory, The Robert M. Buchan Department of Mining, Queen's University, Kingston, ON K7L 3N6, Canada; 16gaa3@queensu.ca (G.H.); ahmad.g@queensu.ca (A.G.)
* Correspondence: hm90@queensu.ca

**Abstract:** The significant neutralization of waste streams required after the acidic bio-oxidation of sulfidic gold ores could be avoided by performing a novel treatment at circumneutral pH with an in situ neutralization. For the first time, the white-rot fungus *Phanerochaete chrysosporium* was incubated in a modified culture medium containing corn steep, an industrial waste product, to support microbial activity and, subsequently, the oxidation of a sulfidic ore at an initial circumneutral pH environment. In this investigation, the concentration of the native culture medium ingredients was first evaluated with response surface methodology to attain maximum sulfide oxidation. The statistical analysis proposed a modified culture medium composed of 12.86 g/L glucose, 2.20 g/L malt extract, 1.67 g/L yeast extract, and 0.49 g/L $MgSO_4 \cdot 7H_2O$ to reach a maximum of 28.7% sulfide oxidation after 14 d-bio-oxidation. pH-controlled batch cultures showed that an increase in initial pH in the range of 5.8 to 7.0 reduced the microbial activity, affecting sulfide oxidation. In addition, the modified culture medium at which yeast extract was substituted with 1.67 g/L corn steep produced comparable microbial activity and sulfide oxidation after 14 d, attaining 21.6% at 5% *w/v* with a maximum 39 U/L lignin peroxidase and 116 U/L manganese peroxidase. A 40.6% sulfide oxidation and 43.8% gold recovery were obtained after 42 d three-cycle replenishing bio-oxidation and 24 h cyanidation, respectively. Overall, corn steep waste showed the potential to substitute more expensive culture medium ingredients, supporting microbial activity and oxidation of sulfidic gold ores at an initial circumneutral pH and contributing to circularity of waste management.

**Keywords:** biotechnology; low-grade refractory gold ores; *Phanerochaete chrysosporium*; waste utilization; sulfide removal; waste circularity



## 1. Introduction

The industrial process design, including acidic biological oxidation (bio-oxidation) followed by cyanidation, has shown a high effectivity in dealing with the refractoriness of sulfidic ores and has been implemented in numerous industrial gold (Au) plants across the globe [1]. Nonetheless, the industrial process results in a high amount of waste streams as acidic slurries generated in standard bio-oxidation (pH 1.0–2.0) are subsequently neutralized with slaked lime and limestones before cyanide leaching, leading to a more complex process circuit [2]. In standard bio-oxidation using acidophilic bacteria, pH regulation during the process and succeeding neutralization represents about 30% of the total operational cost [3]. Hence, a bio-oxidation process carried out at circumneutral pH with an in situ neutralization is a new development that could lessen the burden of new waste production, reinforce environmental protection, and simplify the current circuit process.

A bio-oxidation process avoiding extremely acidic pH requires microorganisms, e.g., fungi or bacteria, capable of growing at pH > 4.0. However, the use of these types of microorganisms to catalyze the oxidation of sulfidic ores in biomining has been barely investigated in comparison to acidophilic bacteria [2,4]. Recent investigations have shown

a limited capacity of neutrophilic bacteria to oxidize solid matrices, while fungi have exhibited a more significant potential to assist the oxidation of sulfidic ores [5,6]. The oxidative capacity of the fungus *Phanerochaete chrysosporium* (*P. chrysosporium*) has been chiefly investigated in a pH range of 4.0 to 5.0, catalyzing the oxidation of sulfidic concentrates and pyrite [7,8]. In our previous investigation, a low-grade sulfidic ore was bio-oxidized at a circumneutral pH, showing the capacity of *P. chrysosporium* at this pH range [6]. The fungal action of *P. chrysosporium* is based on the secretion of ligninolytic enzymes, including lignin peroxidase (LiP) and manganese peroxidase (MnP), which are mainly responsible for the alteration of different substrates [9]. The mechanism for sulfide oxidation using *P. chrysosporium* has been proposed [7]; however, microbial oxidation at pH > 4.0 is likely reduced due to the formation of insoluble iron phases such as iron hydroxide (Equation (1)) as well as the exhaustion of microbial activity over time [8,10].

$$FeS_2 + 3.75O_2 + 3.5\,H_2O \rightarrow 2SO_4^{2-} + 4H^+ + Fe(OH)_3 \quad (1)$$

In processes using microorganisms, the culture medium (CM) composition and incubation conditions, e.g., pH, temperature, and agitation, have a key role because they can affect the gene expression for a wide array of enzymes as well as growth and metabolism. In the case of white-rot fungi such as *P. chrysosporium*, it can significantly influence the activity of ligninolytic enzymes and, ultimately, its response to alter substrates [11,12]. A large variety of medium components have been used to culture *P. chrysosporium* and stimulate the secretion of extracellular enzymes such as LiP and MnP, as displayed in Table 1. The enzyme activity for LiP and MnP has ranged between 1 to 1375 U/L and 86 to 2600 U/L, respectively, depending on the type of CM and initial pH. Most frequently, the CM to grow *P. chrysosporium* is conditioned at pH < 5.0 as the fungus has optimal activity in the pH range of 4.0–4.5 [13]. Investigating the effect of CM and optimal culture conditions at pH > 5.0 seems necessary to expand the understanding of this fungus into other applications, taking advantage of its robustness to survive in a wider pH range [14,15].

**Table 1.** Variation in the concentration of extracellular enzyme secreted by *P. chrysosporium* caused by pH and CM [a].

| Initial pH | Final pH | Culture Medium | Time Maximal Enzyme Secretion (d) | Maximal LiP (U/L) | Maximal MnP (U/L) | Sulfide Oxidation (%) | Reference |
|---|---|---|---|---|---|---|---|
| 4.0–5.5 | NR | 5 g/L steam-exploded straw, 1.25 g/L wheat bran, 0.01 g/L tween 80, 0.005 g/L MgSO$_4$, 0.005 g/L sodium glutamate | 5 | 1375 | 2600 | - | [16] |
| 4.8 | NR | 10 g/L glucose, 3 g/L peptone, 1 g/L corn steep, 1 g/L KH$_2$PO$_4$, 0.5 g/L MgSO$_4$·7H$_2$O | 3 | 475 | 610 | - | [17] |
| 4.5 | 4.5–5.0 | 5 g/L glucose, 2 g/L KH$_2$PO$_4$, 0.05 NH$_4$Cl, 0.5 g/L MgSO$_4$·7H$_2$O, 0.1 g/L CaCl$_2$·2H$_2$O, 0.1 µg/L thiamine HCl, 10 mL/L TE | 5 | 234 | 172 | - | [11] |
| 4.5 | NR | 10 g/L dextrose, 0.11 g/L ammonium tartrate, 7 mg/L MnSO$_4$, 0.3 g/L veratryl alcohol, 0.5 g/L tween 80 | 8 | 541 | 86 | - | [12] |

**Table 1.** *Cont.*

| Initial pH | Final pH | Culture Medium | Time Maximal Enzyme Secretion (d) | Maximal LiP (U/L) | Maximal MnP (U/L) | Sulfide Oxidation (%) | Reference |
|---|---|---|---|---|---|---|---|
| 4.0 | NR | 8 g millet/flask, 2 g wheat bran/flask | NR | NR | NR | 57% at 30% *w/v* | [18] |
| 5 | 2.1 | 10 g/L glucose, 0.2 g/L KH$_2$PO$_4$, 1 g/L MgSO$_4$·7H$_2$O, 0.37 g/L ammonium tartrate, 0.02 g/L CaCl$_2$, 1 mg/L thiamin-HCl, and 70 mL/L TE | 8 | 95 | 316 | 33% at 5% *w/v* | [8] |
| 5.7 | NR | 10 g/L glucose, 2 g/L KH$_2$PO$_4$, 0.5 g/L MgSO$_4$·7H$_2$O, 0.1 g/L CaCl$_2$, 0.2 g/L ammonium tartrate, 0.012 g/L yeast extract, 0.5 g/L tween 80, 0.07 g/L veratryl alcohol, 60 mL/L TE | 6 | 1 | 373 | - | [19] |
| 6.0 | 3.0 | 0.5 g/L lignin, 2 g/L KH$_2$PO$_4$, 0.25 g/L MgSO$_4$·7H$_2$O, 0.1 /L CaCl$_2$, 5 mg/L MnSO$_4$, 10 mg/L VB1, 0.2 g/L ammonium tartrate, 150 mL/L TE | 4 | 240 | 300 | - | [14] |
| 6.0 | 6.6 | Swine wastewater containing 1160 mg/L total N; 213 mg/L total P | 4 | 500 | 650 | - | [15] |
| 5.8 | 7.6 | 10 g/L glucose, 3.5 g/L malt extract, 2.5 g/L yeast extract, 0.5 g/L MgSO$_4$·7H$_2$O | 7 | 51 | 128 | 23% at 5% *w/v* | [6] |

[a] Not reported (NR), trace elements (TE).

In most investigations, glucose has been employed as a primary source of carbon (C), while different sources of nitrogen (N), including inorganic and organic substrates such as ammonium chloride, ammonia tartrate, yeast extract, peptone have been tested to trigger the secretion of enzymes under N and C limited conditions [20,21]. The current literature review has evidenced a need for more utilization of waste products as an alternative source of nutrients to grow microorganisms, especially at pH > 5.0. Most CM components are frequently considered costly and unsuitable for large-scale processes. Therefore, replacing standard ingredients with inexpensive options is still a matter of study. Incorporating waste products from different industrial operations, e.g., sugarcane molasses, wood sugars, beer yeast waste, food processing waste, and wastewater, can boost waste circularity management, resulting in more sustainable and viable potential applications [22,23]. The circularity of waste management is an essential contribution of this investigation by evaluating corn steep waste as a cost-effective source of C and N by supplying the microorganism with micro- and macro-nutrients. Proper valorizing organic waste can avoid the accumulation of municipal solid waste and global greenhouse emissions, which are current issues worldwide.

Taking into account the research gap of earlier investigations, this study pertains to examining the effect of CM and the application of corn steep waste as a substitute to support microbial activity and, eventually, the oxidation of a sulfidic ore at an initial circumneutral pH. Compared to previous studies that have focused primarily on a fungal oxidative treatment at acidic pH using conventional reagents (Table 1), the main novelty

of this investigation lies in the assessment of waste to support microbial activity and, subsequently, the oxidation of a sulfidic ore at an initial circumneutral pH.

As shown in Figure 1, the microbial activity and sulfide oxidation were first assessed in the native CM (glucose, malt extract, yeast extract, and $MgSO_4 \cdot 7H_2O$) by modifying the individual concentration of each ingredient. Subsequently, the evaluation of the CM was performed using D-optimal response surface methodology (RSM) using Design Expert 7.0.0 (State-Ease Inc., Minneapolis, MN, USA) to find out modified culture conditions to maximize sulfide oxidation. Then, pH-controlled batch cultures using phosphate buffer were studied, and their effect on sulfide oxidation was determined. Finally, corn steep was assessed in the modified CM and compared with the standard culture conditions concerning enzyme activity, sulfide oxidation, and Au recovery after applying a replenishing bio-oxidation strategy.

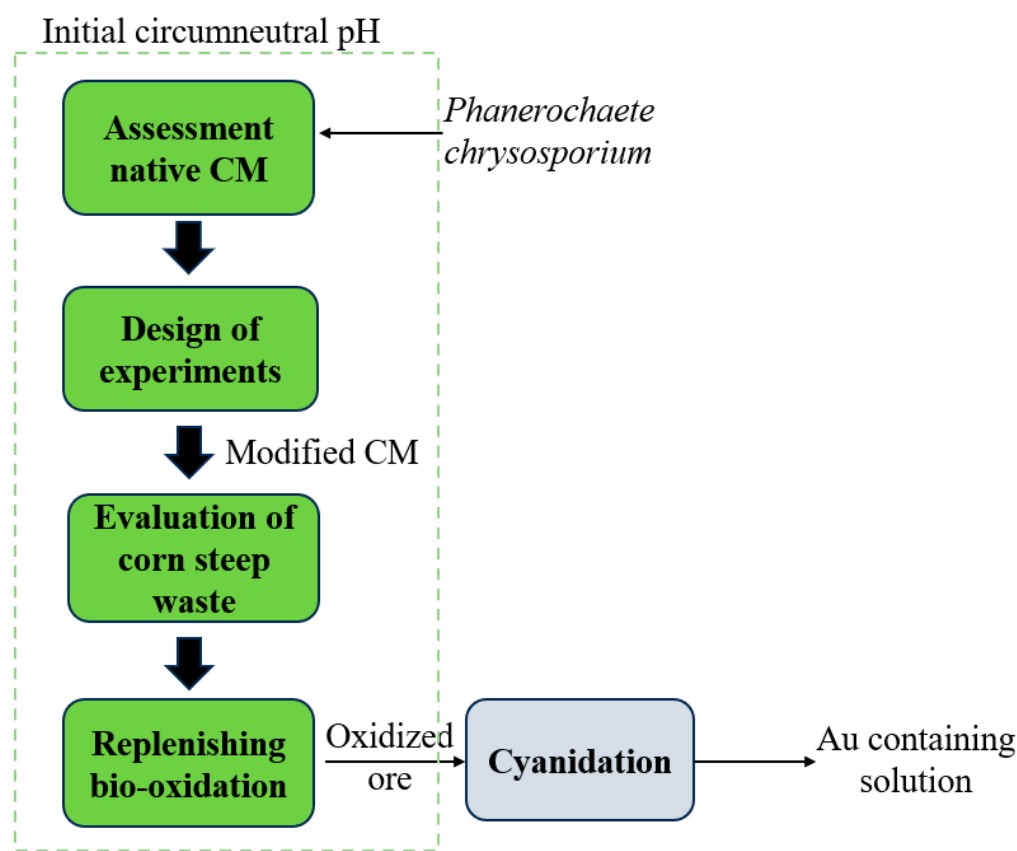

**Figure 1.** Schematic diagram for sulfide bio-oxidation assisted by *P. chrysosporium*.

## 2. Material and Methods

### 2.1. Low-Grade Sulfidic Ore

A sulfidic Au ore from a Canadian deposit classified as a refractory and low-grade source was used in this investigation. In the first instance, the Au-bearing ore was crushed and pulverized until obtaining a particle size < 75 µm for bio-oxidation experiments. Manual rifflers were used to ensure the representativeness of the samples. A laser particle size analyzer (Malvern Mastersizer 3000, Malvern, UK) was employed to determine the 80% passing size of the particles, which was 55 µm (Figure S1). The primary mineralogical characteristics of the samples were determined with X-ray diffraction analysis (XRD) to be pyrite quartz, illite, and magnetite (Figure S1). From Table 2, the ore has an Au grade of 1.69 ppm and 5.7% $w/w$ total sulfur, containing traces of metals such as copper, magnesium, and nickel.

**Table 2.** Elemental composition of the low-grade sulfidic ore.

| Analyte | Ag | Au | Al | Co | Cu | Fe | Mg | Ni | S | V |
|---|---|---|---|---|---|---|---|---|---|---|
| Concentration (ppm) | 6.3 | 1.69 | $5.4 \times 10^3$ | 76 | 661 | $5.0 \times 10^3$ | 2820 | 18 | $5.7 \times 10^3$ | 80 |

### 2.2. Microorganism and CM

The biosafety level one fungus *Phanerochaete chrysosporium* (ATCC 34541) from The American Type Culture Collection (ATCC) was used throughout this investigation. The fungus was first stored in Petri dishes on malt extract agar medium consisting of 15 g/L bacto agar (Sigma-Aldrich, Oakville, ON, Canada) and 30 g/L malt extract (Fisher Scientific, Ottawa, ON, Canada) at 37 °C. Subsequently, a spore suspension having an optical density (OD) of 0.3 respecting water was created by washing the fungus in malt extract-agar medium with sterile deionized water.

The fungal suspension served as inoculum to start the culture of *P. chrysosporium* in the standard CM composed of 10 g/L glucose, 3.5 g/L malt extract, 2.5 g/L yeast extract, and 0.5 g/L MgSO$_4$·7H$_2$O (all analytical grade, Fisher Scientific, Ottawa, ON, Canada). Unless otherwise stated, 1 mL of the spore suspension was inoculated in a total of 100 mL working CM using 250 mL Erlenmeyer flasks covered with a foam stopper. The microorganism was initially incubated stationarily (0 rpm) in an incubator (Biobase; BJPX-2102C, Jinan, China) at 37 °C for 7 d. Before all the experiments, solutions and glassware pieces were sterilized using a 0.2 μm rapid flow filter (Cole-Parmer, Quebec City, QC, Canada) and an autoclave (Tuttnauer; Valueklave 1730, Hauppauge, NY, USA) at 121 °C for at least 20 min, respectively. When required, 0.5 M NaOH was used to adjust the pH solution [15]. Afterward, corn steepin dried format (Sigma-Aldrich) was subsequently used in the CM as an alternative energy source.

### 2.3. Experimental Design of Bio-Oxidation Tests

The bio-oxidation experiments were outlined in six stages, in which the concentration of individual medium ingredients was varied to monitor changes in enzyme activity and sulfide oxidation to optimize the composition of the standard CM. In this way, the enzyme activity was first monitored in the standard CM. Subsequently, the effect of CM modifications on sulfide oxidation and final pH was determined. Next, the standard CM was modified to maximize sulfide oxidation in the sulfidic ore using D-optimal RSM. Afterward, pH-controlled batch cultures with phosphate buffer (6.8 g/L KH$_2$PO$_4$ adjusted to specific pH with 0.2 M NaOH) were used to maintain the pH constant throughout the experiments [24]. Then, the replacement of standard medium components by corn steep, a waste product from food processing industries, was assessed regarding enzyme activity, sulfide oxidation, and pH trends. Finally, experiments were scaled up in a 2 L reactor, and sulfide oxidation and Au recovery were evaluated employing a replenishing strategy for three cycles (14 d bio-oxidation per cycle). In replenishing tests, once a bio-oxidation cycle was finished, the residual spent liquid was replaced with a new microbial culture, continuing with the treatment for the next cycle [6].

The bio-oxidation experiments were all started at pH 5.8 using an inoculum of 1% (*v/v*) fungal spores in fresh CM under a two-step approach. Hence, the fungal culture was allowed to grow steadily at 0 rpm and 37 °C in a temperature-controlled incubator prior to commencing the experiments with the sulfidic ore. After 7 d of growth, enzyme activity, pH, and oxidation-reduction potential (ORP) were recorded. The choice of 7 d for the pre-growth stage is based on the time required to reach robust microbial growth, which is frequently observed after 5 d to 8 d of incubation [12,18]. In order to maintain consistency and comparison with our previous investigation, 7 d of incubation was maintained in this research [6]. After the growing stage, bio-oxidation tests were begun using a 5% *w/v* PD (pulp density) sulfidic ore, setting the agitation speed at 150 rpm for 14 d. Control runs were also carried out using identical experimental conditions, excluding microbial

inoculation. All the experiments were performed at least in duplicates in this investigation, presented as mean values ± standard deviation.

### 2.4. Enzyme Assays

The enzyme activity of LiP and MnP was determined by spectrophotometrically measuring the absorbance of the culture filtrate under specific assay conditions using the oxidation of veratryl alcohol (310 nm) and $Mn^{2+}$ (240 nm), correspondingly [6,25]. One unit (U) of a particular enzyme activity is designated as the amount of enzyme needed to form 1 μmol of its specific product per min per liter (U/L). For LiP, the activity was quantified following the formation of veratraldehyde in a 3.0 mL solution composed of 100 mM sodium tartrate (pH 3.0), 10 mM of veratryl alcohol, 10 mM $H_2O_2$, and 0.4 mL culture filtrate. In the case of MnP, the enzymatic activity was monitored considering the formation of $Mn^{3+}$ in a 3 mL solution comprising 50 mM sodium succinate (pH 4.5), 15 mM of $MnSO_4$, and 10 mM $H_2O_2$, and 0.4 mL culture filtrate.

### 2.5. Statistical Analysis of CM

Following the determination of the influential concentration for each component in the CM, the optimization of sulfide oxidation was investigated on the basis of the design of experiments (DOE) using D-optimal RSM in the software Design Expert 7.0.0. DOE is a satisfactory methodology that offers a quick alternative to reduce the number of experimental runs with high precision. This statistical analysis also includes the interaction and numeral relationships among the considered relevant parameters. In order to reduce the number of experiments and maximize sulfide oxidation of the sulfidic ore, 25 runs were designed by D-optimal RSM using the selected levels for each parameter, as shown in Table S1.

The experimental designation shown in Table S1 is prepared based on the relevant concentration of culture medium ingredients, that is, glucose (10–20 g/L), malt extract (0–3.5 g/L), yeast extract (0–2.5 g/L), and $MgSO_4 \cdot 7H_2O$ (0–0.5 g/L) as well as D-optimal DSM.

After finding the modified concentration for each component for maximal sulfide oxidation based on analysis of variance (ANOVA), replicates were performed to confirm the result suggested using the software Design of Expert 7.0.0.

### 2.6. Analytical Techniques

The as-received ore sample was characterized with coupled plasma–optical emission spectrometry (PerkinElmer, Waltham, MA, USA) and XRD (Philips, Eindhoven, The Netherlands). Ultraviolet-visible spectrophotometry (Thermo Scientific GENESYS 10S, Ottawa, ON, Canada) was used to determine enzyme activity secreted by *P. chrysosporium* and the OD of the fungal spore concentration used as inoculum in all the experiments. A pH/ORP meter with an Ag/AgCl electrode (VWR) was utilized to monitor the pH and ORP of filtrated liquid samples. The content of sulfur in all the solid samples was determined with a carbon-sulfur analyzer (Eltra CS 2000, Newtown, PA, USA), and the sulfide oxidation was calculated according to Equation (2) [26]. Before determining the total sulfide oxidation, the oxidized sample was previously subjected to three washing steps to eliminate the remaining medium component and biomass, elemental sulfur (S°) and sulfate ($SO_4^{2-}$), following standard methods described in our previous research [6].

$$S_{\text{sulfide oxidation}} = \frac{W_I S_I - W_F W_F}{W_I S_I} \times 100 \tag{2}$$

where $W_I$ and $W_F$ designate the weight (g) of the initial sample and final residue, respectively. $S_I$ and $S_F$ are the initial and final sulfur content (% $w/w$), respectively.

After performing conventional cyanidation in the samples (2000 ppm NaCN, 30% $w/v$ PD, 24 h, pH 11), Au recovery was determined indirectly by considering the difference between the assay of Au in the head sample and the oxidized sample using Au fire assay.

## 3. Results and Discussion

### 3.1. Effect of CM on Enzyme Concentration

Enzyme activity in response to variations in the concentration of each ingredient in the standard CM is displayed in Figure 2. According to the results, there was a negligible influence of the concentration of $MgSO_4 \cdot 7H_2O$ in the range of 0–1 g/L, whilst the variation in concentration of the rest of the medium components was more significant during the growth of *P. chrysosporium*. From Figure 2a, an increasing concentration of glucose up to 10 g/L boosted the secretion of extracellular enzymes, attaining 51 U/L LiP and 128 U/L MnP. However, the availability of greater concentrations apparently depressed microbial activity. The secretion of enzymes by *P. chrysosporium* is denominated as secondary metabolism, depending on nutrient limitation conditions [27]. Glucose, a conventional carbohydrate used in CM, is categorized as a simple sugar (monosaccharide), serving as a rich C source for microorganisms to use as a growth substrate [21]. As the primary source of C in the selected CM, glucose was essential in the secretion of enzymes in the 0–10 g/L range, requiring at least 10 g/L to obtain a better performance.

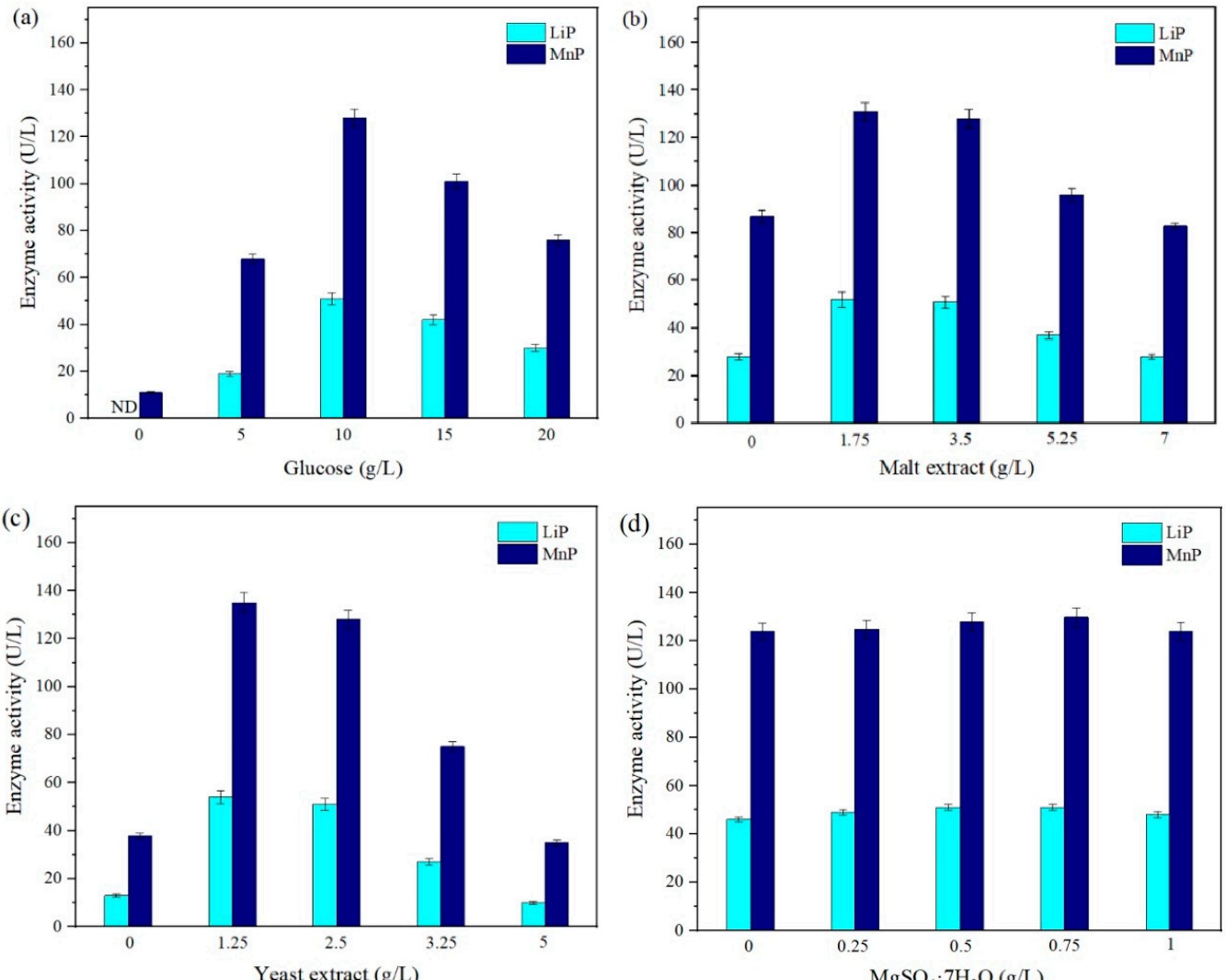

**Figure 2.** Effect of (**a**) glucose, (**b**) malt extract, (**c**) yeast extract, and (**d**) $MgSO_4 \cdot 7H_2O$ on enzyme activity (U/L) secreted by *P. chrysosporium* after 7 d of incubation at 0 rpm and 37 °C. Not detected (ND).

Notably, glucose removal significantly affected the enzyme activity of *P. chrysosporium*, resulting in 0 U/L LiP and 11 U/L MnP, which could be ascribed to the unavailability of simple C sources to support microbial activity [27,28]. From Figure 2a, *P. chrysosporium* could likely require a glucose concentration higher than 5 g/L to attain an optimal fungal metabolism. A supply of 5 g/L glucose in the CM could have also experienced an earlier development of enzymes due to the faster glucose consumption. Thus, the onset of enzymes could have occurred before 7 d, showing a lower concentration on day 7. In the same way, concentrations higher than 10 g/L glucose could have undergone a later development of enzyme activity having an onset after day 7 and, therefore leading to a lower concentration of enzymes at day 7 of incubation. Concentrations exceeding 15 g/L displayed detrimental influence as white-rot fungi such as *P. chrysosporium* might require a C-limited environment to favor the secretion of extracellular enzymes, which turned out in 30 U/L LiP and 76 U/L MnP.

Based on Figure 2b, the optimal concentration of malt extract was in the range of 1.75 to 3.5 g/L, reaching a maximum of 52 U/L LiP and 131 U/L MnP at a concentration of 1.75 g/L malt extract. Apparently, any increase or decrease outside this range of concentration had a negative impact on the secretion of LiP and MnP. Malt extract, a compound derived from malted barley, is mainly a source of carbohydrates encompassing reduced sugars (>90%) such as maltose, fructose, and sucrose and also has a smaller proportion of N constituents such as amino acids, peptides, and minerals [29,30]. A higher concentration of over 3.5 g/L of this component could have created nutrient-rich conditions, affecting the secondary metabolism of the microorganism. Further, the ideal behavior of the fungi was also affected by the removal of malt extract (0 g/L) but was probably supported by other nourishing medium components such as yeast extract and glucose, resulting in a concentration of 28 U/L LiP and 87 U/L MnP.

Comparably to malt extract, a concentration of yeast extract in the range of 1.25–2.5 g/L was more favorable for enzyme secretion, attaining a maximum of 54 U/L LiP and 135 U/L MnP, as shown in Figure 2c. A concentration over this range was ineffective in improving the microbial activity of *P. chrysosporium*, probably due to excess nutrients in the CM. In addition, the elimination of yeast extract (0 g/L) caused a sharp decrease in enzyme concentration (13 U/L LiP and 38 U/L MnP), highlighting the importance of this medium component in the CM. Yeast extract is a nourishing compound and is one of the primary sources of N, vitamins (biotin and vitamin B), and TE, e.g., manganese (Mn), magnesium (Mg), calcium (Ca), phosphorus (P), Zinc (Zn), iron (Fe), Cobalt (Co), as well as proteins (mainly nucleic acid and glutathione) and amino acids required for the metabolism, growth, and better function of this microorganism [31]. Therefore, its composition makes yeast extract a highly indispensable ingredient in the CM. Contrary to the rest of the medium ingredients, there was an insignificant influence of the concentration of $MgSO_4 \cdot 7H_2O$ in the range of 0–1 g/L, resulting in similar concentrations of enzymes at different concentrations, which could be due to the supplementation of rich ingredients in the standard CM that have TE of a wide variety of minerals and nutrients, e.g., Mg, Mn.

### 3.2. Effect of CM on Sulfide Bio-Oxidation

Figure 3 displays the changes observed in sulfide oxidation and pH solution after 14 d of contact time between *P. chrysosporium* and the sulfidic ore. From Figure 3, it was deduced that sulfide oxidation varied from nearly 1.2% to 25.2%, considering individual changes in the concentration of medium components, which generally coincided with the trends in enzyme activity. The mechanism of the white-rot fungi to alter reduced sulfur species depends on the secretion of extracellular enzymes, including LiP and MnP. The catalytic cycle of oxidative enzymes has been explained in previous publications [4,6] and essentially consists of a chain of oxidation-reduction reactions favoring the oxidation of reduced sulfur species and the reduction of enzymatic compounds secreted by *P. chrysosporium*. However, the mechanism still needs further research to fully understand this type of system.

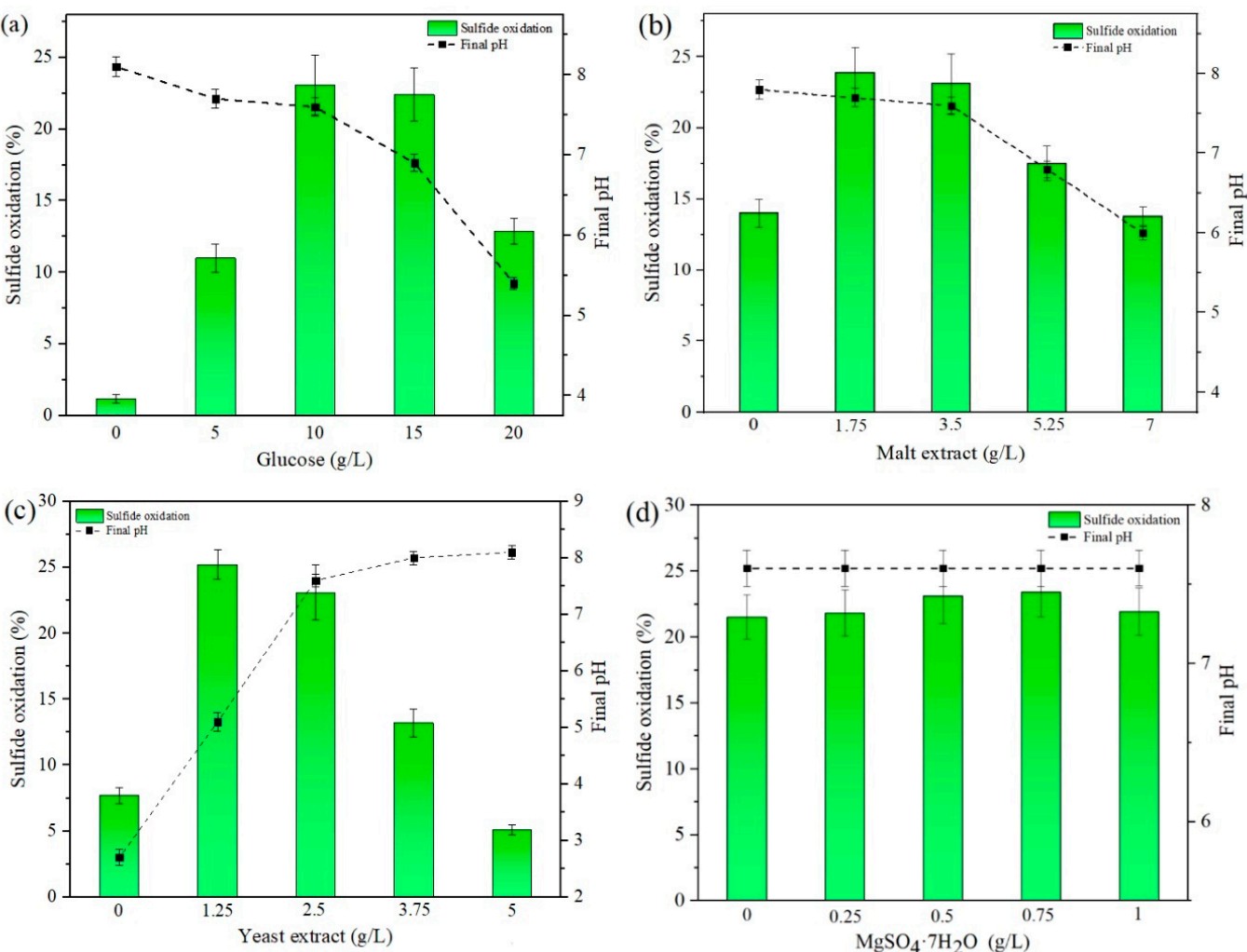

**Figure 3.** Effect of (**a**) glucose, (**b**) malt extract, (**c**) yeast extract, and (**d**) MgSO$_4$·7H$_2$O on sulfide oxidation and final pH in 14-d bio-oxidation experiments using *P. chrysosporium* at 5% *w/v* PD and 37 °C.

Concerning the experiments with variations in glucose concentration (Figure 3a), a concentration between 10 to 15 g/L resulted in ca. 23% oxidation, whilst 0 and 20 g/L reduced the capacity to oxidize the sulfidic ore, generating 1.2% and 12.9%, correspondently.

Furthermore, 1.75–3.5 g/L malt extract and 1.25–2.5 g/L yeast extract led to better sulfide oxidation compared to other concentrations of these reagents, displaying a maximum of 23.9% and 25.2%, correspondingly (Figure 3b,c). A concentration out of these two ranges of concentration led to a decrease in sulfide oxidation, reaching 13.8% and 5.1% at 20 g/L malt extract and yeast extract, respectively. The addition of different concentrations of MgSO$_4$·7H$_2$O in the interval of 0–1 g/L had an insignificant influence compared to other parameters, such as yeast extract and glucose, resulting in similar oxidation (ca. 23%) for all the evaluated conditions.

Sulfide oxidation increased proportionally to glucose concentration in the CM up to the range of 10 to 15 g/L, and subsequently, sulfide oxidation decreased at higher concentrations. This trend is an outcome of two coexisting factors: (1) glucose, a simple source of C, serves as the primary source of C supporting the microbial activity during the initial growth, and (2) the availability of higher concentration of glucose could prevent the secretion of ligninolytic enzymes under nutrient-rich cultures [27,28]. The higher sulfide oxidation observed in the concentration range of 1.75–3.5 g/L malt extract and 1.25–2.5 g/L yeast extract could be associated with the fact that a nutrient deficiency, but sufficiently enough nutrients to support the fungal activity benefits the microbial activity secreting

a higher concentration of enzymes and therefore better sulfide oxidation [20,21]. The effect of $MgSO_4 \cdot 7H_2O$ on sulfide oxidation could be interpreted differently compared to other reagents in the CM because it had a negligible impact on ligninolytic enzyme secretion (Figure 2d) and sulfide oxidation (Figure 3d) in the evaluated interval. Even though the concentration of Mg has been reported to favor fungal growth [32], it seems that the concentration of yeast extract and malt extract in the evaluated CM could have been sufficient to provide the Mg content and other TE required for fungal growth [30,31].

The initial pH for all the tested conditions was previously adjusted to pH 5.8 before starting the tests. However, major pH changes were measured after 14 d of bio-oxidation, as displayed in Figure 3. Results showed that the final pH fluctuated from 2.7 to 8.1, depending on the concentrations of each reagent in the CM. A more notable decrease in pH was recorded when yeast extract was removed entirely from the CM (0 g/L), resulting in pH 2.7. By adding yeast extract to the CM, the pH changes could be attributed to the buffering properties of this compound and microbial activity in the CM [33,34]. The pH trends were similar for glucose and malt extract; a rise in their concentration produced a pH drop more remarkably. Glucose and malt extract are metabolized by white-rot fungi, releasing organic acids and protons into the solution and modifying pH [34,35]. Therefore, an increase in glucose and malt extract concentration seems to have caused a higher organic acid concentration and a more significant drop in the pH solution, recording 5.4 and 6.0 at 20 g/L and 7 g/L, respectively. Regarding the effect of different concentrations of $MgSO_4 \cdot 7H_2O$ on pH changes, the final pH remained at ca. 7.6 for all the evaluated concentrations, which seems to be reasonable based on the similarity in sulfide oxidation found in Figure 3d.

### 3.3. Effect of Interaction of Medium Components on Sulfide Oxidation

After analyzing the role of each component in the standard CM, a set of tests was suggested to optimize the bio-oxidation of the sulfidic ore. The experiments were proposed based on D-optimal RSM design, including 25 experiments displayed in Table S1 (supplementary material) to obtain modified combinations of ingredients at different concentrations for more efficient bio-oxidation. The design contemplated the influential concentration of glucose (10–20 g/L), malt extract (0–3.5 g/L), yeast extract (0–2.5 g/L), and $MgSO_4 \cdot 7H_2O$ (0–0.5 g/L) found in the preliminary study. The design provides an overview of the relevance of individual parameters and their interactions, resulting in a model to predict bio-oxidation. The software outcomes are presented in diagrams of sulfide oxidation versus medium ingredients and ANOVA chart.

Table 3 shows a summary of the statistical data obtained from the multiple linear regression model. In order to guarantee a model reliability within 95% of the confidence, a *p*-value (probability value) of the model is required to be less than 0.05 (F-value higher than 4) with a lack of fit *p*-value greater than 0.05 [36]. The model *p*-value of <0.0001 indicates that it is significant enough and that there is a chance of only 0.01% that the model could take place because of noise. It is also desirable for a model to have a non-significant lack of fit. The lack of fit *p*-value of the model resulted in 0.77, implying that it is not significant to pure error. Table 3 also displays that most of the individual component concentrations were effective terms, as well as some interactions, such as glucose–malt extract. According to the simulated and experimental data analyzed using the software design expert, a quadratic model was proposed with an $R^2$ of 0.92 (Table S2 in supplementary material). The modeled trends provided using the software concur with the experimental data and the prediction of the role of each evaluated ingredient concerning sulfide oxidation. The interactions of different parameters over bio-oxidation experiments are shown in Figure 4.

**Table 3.** ANOVA chart for the multiple linear regression model for sulfide bio-oxidation.

| Source | Sum of Squares | df [a] | Mean Square | F-Value | p-Value Prob > F | |
|---|---|---|---|---|---|---|
| Model | 1911.00 | 10 | 191.10 | 15.71 | <0.0001 | significant |
| A-Glucose | 85.27 | 1 | 85.27 | 7.01 | 0.0191 | |
| B-Malt extract | 30.71 | 1 | 30.71 | 2.52 | 0.1345 | |
| C-Yeast extract | 674.16 | 1 | 674.16 | 55.41 | <0.0001 | |
| D-$MgSO_4 \cdot 7H_2O$ | 68.39 | 1 | 68.39 | 5.62 | 0.0326 | |
| AB | 84.78 | 1 | 84.78 | 6.97 | 0.0194 | |
| AD | 9.73 | 1 | 9.73 | 0.80 | 0.3863 | |
| BC | 0.87 | 1 | 0.87 | 0.071 | 0.7933 | |
| $A^2$ | 48.85 | 1 | 48.85 | 4.01 | 0.0648 | |
| $B^2$ | 67.41 | 1 | 67.41 | 5.54 | 0.0337 | |
| $C^2$ | 324.80 | 1 | 324.80 | 26.70 | <0.0001 | |
| Residual | 170.34 | 14 | 12.17 | | | |
| Lack of Fit | 87.16 | 9 | 9.68 | 0.58 | 0.7736 | not significant |
| Pure Error | 83.18 | 5 | 16.64 | | | |
| Cor Total | 2081.34 | 24 | | | | |

[a] degree of freedom.

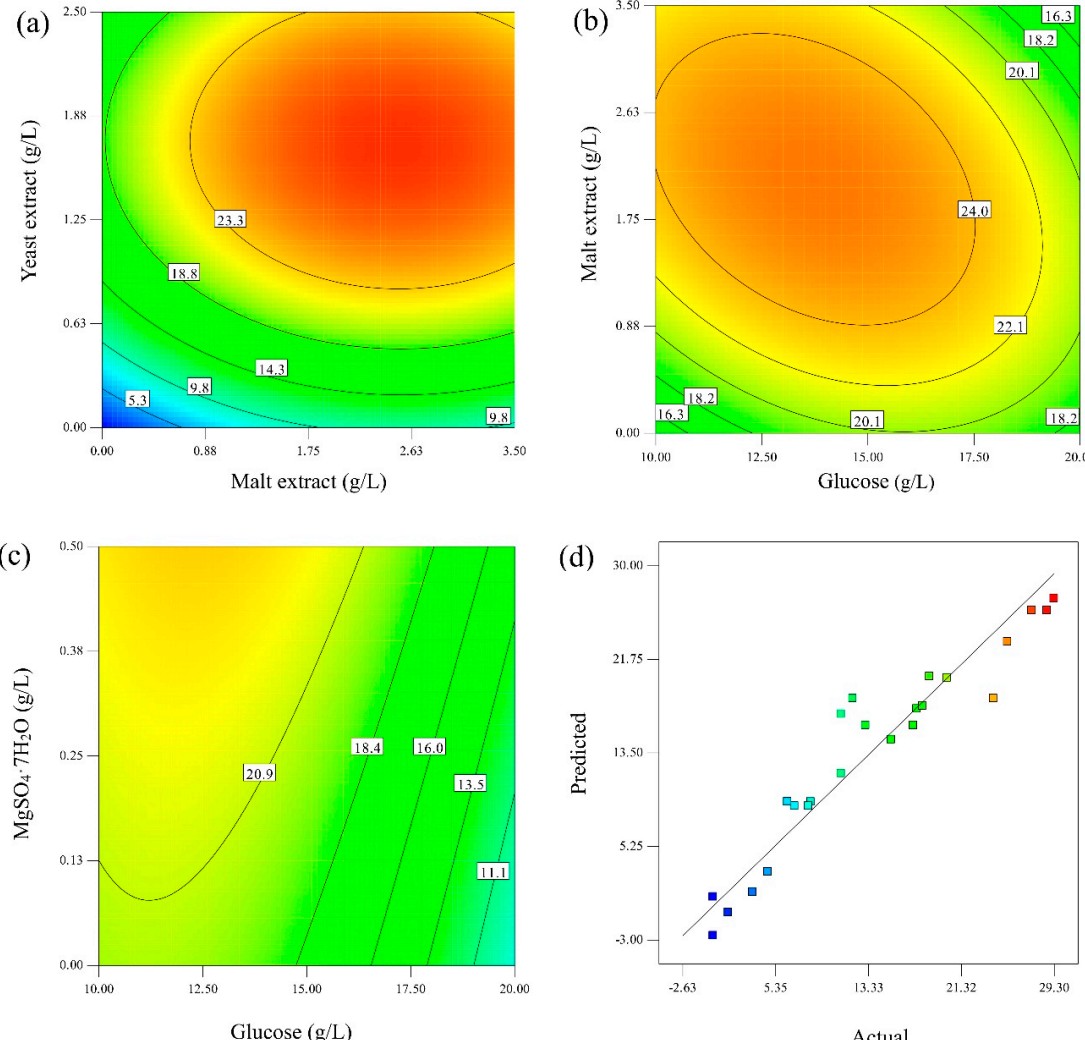

**Figure 4.** Sulfide oxidation as a function of variation in concentration of (**a**) malt extract—yeast extract, (**b**) glucose—malt extract, and (**c**) glucose—$MgSO_4 \cdot 7H_2O$ during 14-d bio-oxidation assisted by *P. chrysosporium* and (**d**) predicted vs. actual value of bio-oxidation using 5% *w/v* PD and 37 °C.

Using a concentration of yeast extract <1 g/L, the concentration of malt extract had a negligible effect on enhancing the sulfide oxidation rate (Figure 4a). Additionally, as both variables approached close to 0 g/L, a considerable reduction in sulfide oxidation was observed, demonstrating the importance of these components in the CM to sustain the microbial activity of the fungus. Variations in the concentration of malt extract became a more influential factor at a yeast extract concentration >1 g/L. If a concentration of 1.70 g/L yeast extract was provided, a change in malt extract concentration in the range of 0.80 to 3.5 was more significant in sulfide oxidation, reaching a maximum of 27.8% sulfide oxidation at 2.35 g/L of malt extract followed by a slight drop at a concentration higher than 3.5 g/L malt extract (25.6% sulfide oxidation). Yeast extract above >2.1 g/L produces a drop in sulfide oxidation, probably due to the nutrient-limited condition required by the fungus to produce more extracellular enzymes.

Regarding the interaction between glucose and malt extract shown in Figure 4b, a concentration of malt extract in the range of 1.0–2.9 g/L resulted in >24% sulfide oxidation using a glucose concentration between 12 to 15 g/L. By keeping a concentration of malt extract at 3.0 g/L, a higher concentration of glucose (>15 g/L) seems detrimental, which could be explained by the excess of nutrients in the system. Malt extract is also an N source in a smaller proportion than yeast extract. Figure 4b also highlights the importance of having a lower concentration of N sources to support the microbial activity of this microorganism via an N-deficient environment [27].

Concerning glucose and $MgSO_4 \cdot 7H_2O$ interaction in Figure 4c, $MgSO_4 \cdot 7H_2O$ in the range 0–0.5 g/L was determined to have a negligible effect using 10–15 g/L glucose, as mentioned in the previous section. From Figure 4d, a good correlation was found between the predicted and actual data, which is indicated by the clustering of points around the diagonal, satisfactorily verifying the model's robustness. Based on the analysis performed with D-optimal RSM, Equation (3) was suggested to best fit the sulfide oxidation prediction as a function of the medium ingredients with an $R^2$ of 0.92.

$$\text{Sulfide oxidation (\%)} = -29.44 + 4.25 C_{\text{glucose}} + 10.65 C_{\text{malt extract}} + 20.63 C_{\text{yeast extract}} - 2.89 C_{\text{MgSO4·7H2O}} - 0.30 C_{\text{glucose}} \times C_{\text{malt extract}} + 0.73 C_{\text{glucose}} \times C_{\text{MgSO4·7H2O}} - 0.12 C_{\text{malt extract}} \times C_{\text{yeast extract}} - 0.14 C^2_{\text{glucose}} - 0.145 C^2_{\text{malt extract}} - 5.96 C^2_{\text{yeast extract}} \ (R^2 = 0.92)$$ (3)

where C represents the concentration of each medium ingredient in g/L.

The value of $R^2 = 0.92$ suggests that the sulfide oxidation equation effectively characterizes the system using the tested experimental conditions. Results reveal that maximum sulfide oxidation could be attained using a CM composed of 12.86 g/L glucose, 2.20 g/L malt extract, 1.67 g/L yeast extract, and 0.49 g/L $MgSO_4 \cdot 7H_2O$. The enzyme activity for LiP and MnP in the modified CM resulted in 58 ± 3 and 143 ± 5 U/L, respectively, after 7 d of incubation. Three confirmation experiments were conducted, and a sulfide oxidation of 28.7 ± 1.6% was reached after 14-d bio-oxidation with a final pH of 6.1 ± 0.2. In our previous research work employing the standard CM (10 g/L glucose, 3.5 g/L malt extract, 2.5 g/L, and 0.5 g/L $MgSO_4 \cdot 7H_2O$) with the same ore [6], sulfide bio-oxidation resulted in 23.1 ± 2.1% with final pH 7.6 ± 0.1 after 14 d. The maximum enzyme activity recorded using the standard CM was 51 ± 3 U/L LiP and 128 ± 4 U/L. Hence, the higher oxidation of the sulfidic ores could be explained by the increase in metabolites likely triggered by a nutrient-deficient CM.

A control test, utilizing the modified CM at identical experimental conditions except for the microbial inoculation, was also carried out to evaluate the impact of the CM on sulfide oxidation. The control tests using the modified CM resulted in 0.6 ± 0.2% sulfide oxidation, which could be associated with the chemical oxidation of sulfide, being negligible compared to the fungi-assisted experiments.

The same fungus has been used to oxidize sulfidic ores in an acidic pH environment. Ofori-Sarpong et al. [7] tested *P. chrysosporium* to oxidize 30% *w/v* sulfidic concentrate at an initial pH of 4.0, attaining 57% sulfide oxidation. In addition, Yang et al. [8] also employed the microorganism to oxidize 5% *w/v* of almost pure pyrite at an initial pH of 5.0, reporting

a maximum of 95 U/L LiP and 316 MnP with a 33% sulfide oxidation and a final pH of 2.1. Compared to previous studies, the performance of *P. chrysosporium* appears to be depleted at an initial circumneutral pH using the standard CM and the proposed modified CM. Based on Table 1, there was generally a decrease in microbial activity when comparing different CM at initial acidic and circumneutral pH. In our previous study [6], a pH of 4.0 led to a higher concentration of metabolites than a pH of 7.0, which helped obtain a higher sulfide oxidation under similar conditions. An initial pH of 4.0 seems optimal to favor robust microbial growth; nonetheless, this experimental condition did not lead to a final neutral pH after bio-oxidation.

*3.4. Effect of pH-Controlled Batch Cultures*

After determining the modified CM to maximize sulfide oxidation, experiments were carried out using pH-controlled batch culture to maintain a circumneutral pH throughout bio-oxidation. In our previous study, the cultures with no pH control experienced a decrease in pH to near pH 4.0 once the fungus reached a robust growth after 7 d of incubation, which coincided with the onset of the maximal secretion of ligninolytic enzymes [6]. Subsequently, pH showed an increasing trend as the bio-oxidation progressed over time, resulting in a final pH > 7.0, which could be mainly attributed to the consumption of organic acid buffer capacity of medium components as a result of fungal activity [6,35].

In this section, *P. chrysosporium* was tested at pH 5.8, 6.0, 6.5, and 7.0, maintaining the pH solution with phosphate buffer to avoid a significant change in pH throughout the experiments. Figure 5 compares the data for enzyme activity and sulfide oxidation using the standard and modified CM under pH-controlled and no pH-control conditions. From Figure 5a, trends observed in the modified and standard CM were mainly similar for microbial activity. However, as previously reported, the modified CM seems to favor a higher release of enzymes, likely due to a nutrient-deficient condition [27]. The effect of pH-controlled batch cultures is also evidenced in Figure 5a, showing that the higher the pH of the CM adjusted in the range of 5.8 to 7.0, the lower the microbial activity of *P. chrysosporium*. A pH range of 4.0–4.5 has been reported to be more favorable for *P. chrysosporium* to secrete ligninolytic enzymes [13,37]. Therefore, it is reasonable that higher pH values beyond pH 4.0 slowed down the enzyme activity, as indicated in Figure 5a.

In the modified CM, a higher enzyme concentration was attained using a CM with no pH control, which led to 58 U/L LiP and 143 U/L MnP, followed by the controlled-pH culture adjusted to pH 5.8 with 34 U/L LiP and 78 U/L MnP. A pH-controlled culture conditioned at pH 7.0 likely resulted in the lowest enzyme activity, as enzymes were not detected on day 7 of incubation. In the standard CM, a similar trend was found with the highest enzyme activity of 51 U/L LiP and 128 U/L MnP using no pH control. The lower concentration of enzymes with increasing pH is presumably attributed to the fact that a pH environment outside of the range of 4.0–4.5 might have affected enzyme stability, remarkably leading to a decreased concentration of enzymes [38]. pH is crucial for the stability and activity of heme-containing enzymes such as LiP and MnP, affecting their ionic form of the active sites and, consequently, the binding with the heme group [39]. Another factor to consider is the higher concentration of phosphate salts over 4 g/L used to maintain a pH > 6.0, as an excessive concentration of phosphate salts could have limited the optimal growth and metabolism [40].

Considering Figure 5b, the comparison of sulfide oxidation at different pH-controlled environments reveals that pH-controlled cultures at pH > 5.8 induced a lower sulfide oxidation compared with no control experiments, which principally seems to be interconnected with the reduced activity of enzymes at elevated pHs. Using the modified CM conditions, 28.7% sulfide oxidation could be achieved by utilizing no pH control. The second-best sulfide oxidation was 14.3%, attained in a pH-controlled culture at pH 5.8, followed by 13.2% adjusted the culture to pH 6.0. Successively, the higher pH adjustment of the CM led to lower fungal activity and, consequently, lower sulfide oxidation, reaching only 5.1% and 1.6% while using the

modified CM at pH 6.5 and pH 7.0, respectively. Sulfide oxidation in standard CM followed a comparable tendency at increasing pH values, as shown in Figure 5b.

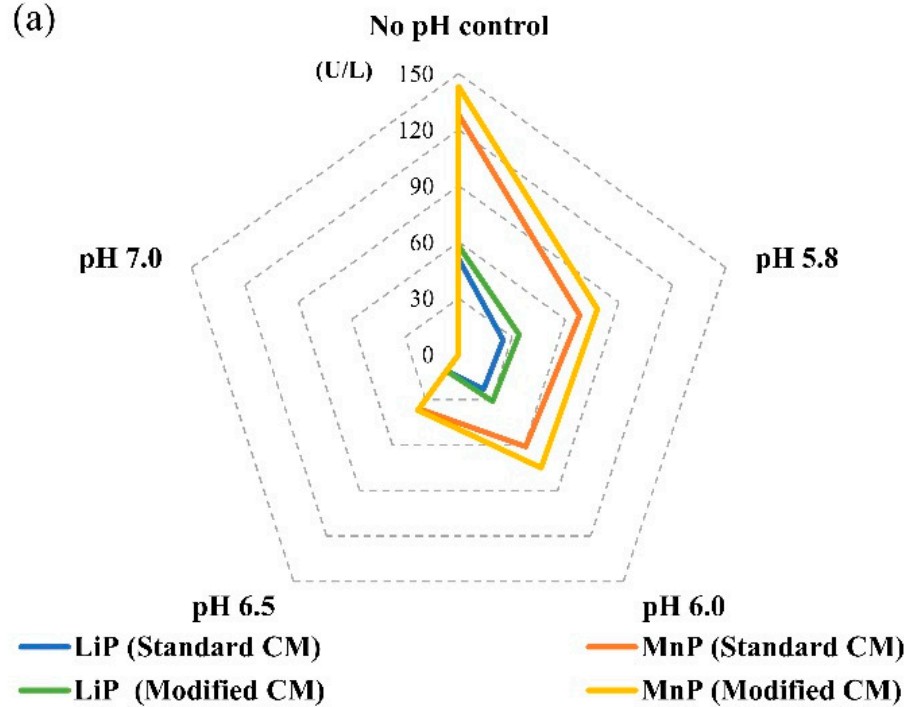

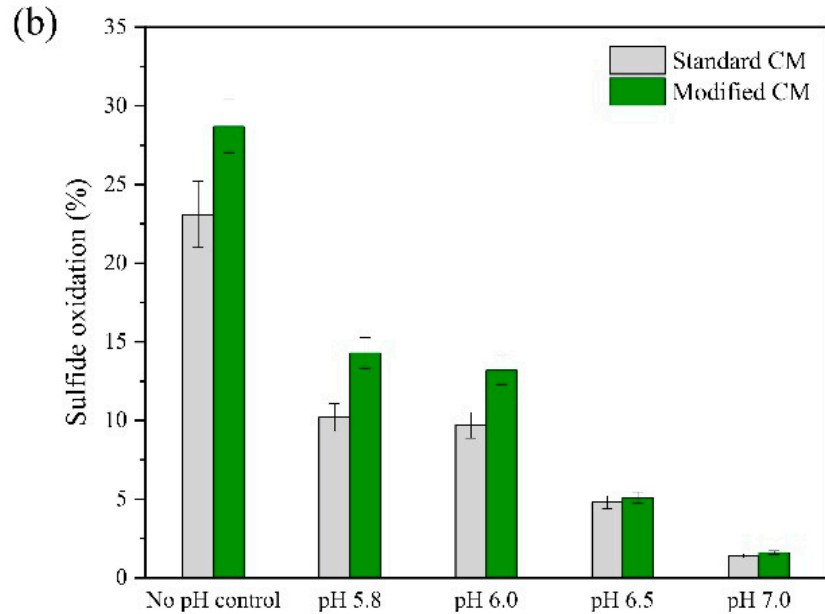

**Figure 5.** (**a**) Enzyme activity (U/L) and (**b**) sulfide oxidation (%) using modified and standard CM in pH-controlled batch cultures with a 5% *w/v* PD and 37 °C.

## 3.5. Evaluation of Corn Steep as a CM Component

Taking into consideration that the proposed investigated CM resulted in partial sulfide oxidation after 14 d, corn steep, a waste medium ingredient, was used as an alternative to increasing the viability of the process from an economic perspective. Thus, the addition of corn steep in the standard CM was investigated as a possible replacement for other nourishing

components, including yeast extract and malt extract. Adding corn steep changed the initial pH solution of the CM to moderately acidic pH (pH 4). Therefore, the pH of all solutions was adjusted to pH 5.8 before inoculation for better comparison with the previous results, as *P. chrysosporium* responds differently to different pH environments [13,28].

Figure 6 shows the effect of corn steep in the standard CM in terms of enzyme activity and sulfide oxidation by replacing the corresponding concentration of malt extract (0–7 g/L) and yeast extract (0–5 g/L). As shown in Figure 6a, the substitution of malt extract for corn steep impacted microbial activity and sulfide oxidation, showing a decreasing trend for higher concentrations of corn steep in the CM in the range of 0–7 g /L. By replacing yeast extract for corn steep (Figure 6b), the addition of corn steep displayed a good contribution to substituting yeast extract at increasing concentrations up to 2.5 g/L, attaining a maximum of 18.2% sulfide oxidation and 33 U/L LiP and 98 U/L MnP. Concentrations above 2.5 g/L seem to affect enzyme secretion and sulfide oxidation, leading to 9 U/L LiP and 29 U/L MnP with 4.1% sulfide oxidation with 5 g/L corn steep in the CM. The final pH in all culture media having corn steep was close to pH 4, which seems to be attributed to the composition of corn steep based on lactic acid [41].

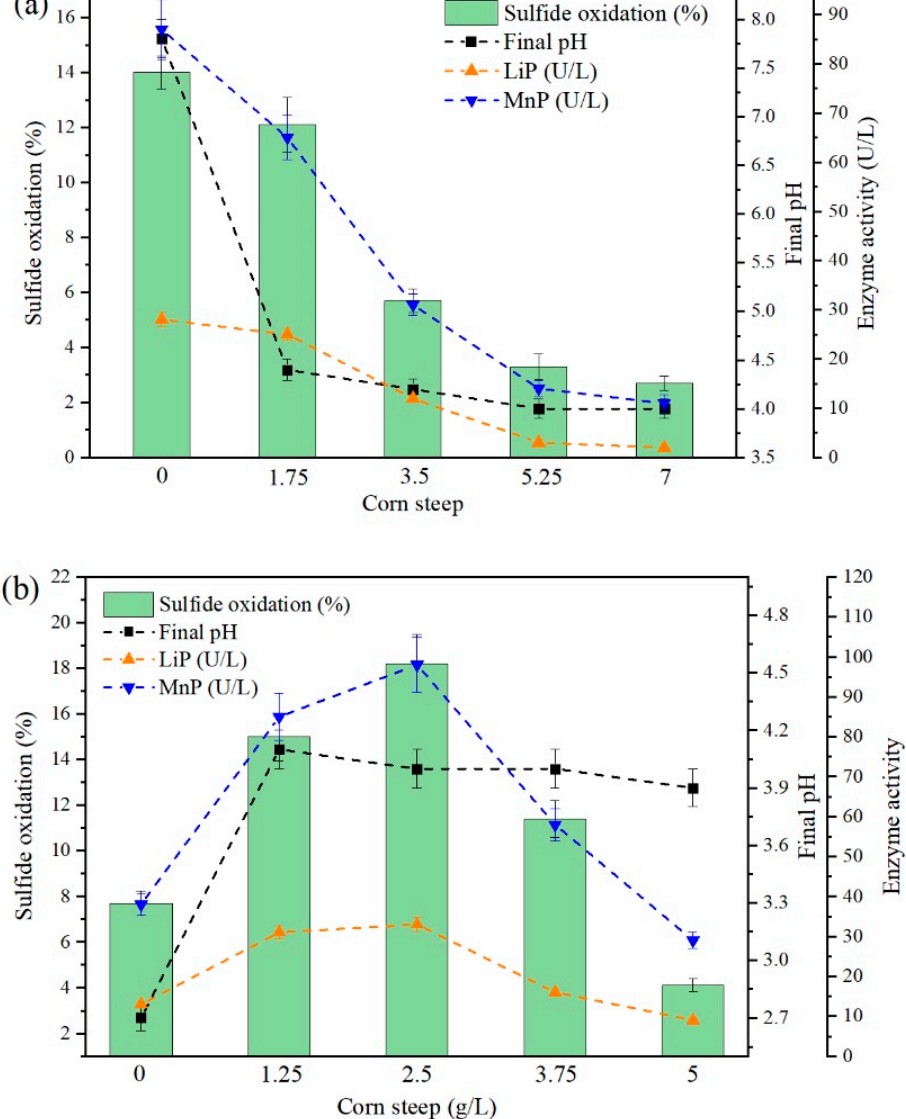

**Figure 6.** Replacement of (**a**) malt extract and (**b**) yeast extract for corn steep in CM and its effect in sulfide oxidation, final pH, and enzyme activity in 14-d bio-oxidation at 5% *w/v* PD.

In the cells of *P. chrysosporium* and other white-rot fungi, a wide variety of metabolic pathways take place simultaneously, e.g., glycolysis, pentose-phosphate pathway, tricarboxylic acid (TCA) cycle, which are all essential for the growth and functionality of fungal cells. Glycolytic pathways, together with the TCA cycle, produce the oxidation of carbohydrates, fatty acids, and amino acids, forming high-energy molecules, e.g., adenosine triphosphate (ATP) and nicotinamide adenine dinucleotide (NADPH), which support the formation of proteins and act as energy storage for the fungal cells [42]. Figure 7 shows the role of the principal metabolic pathways in *P. chrysosporium* cells. The gene expression for different types of enzymes, including LiP and MnP, as well as the intracellular and extracellular interactions, are highly influenced by the type of growth substrates and environmental conditions to which the microorganism is exposed in the CM [43].

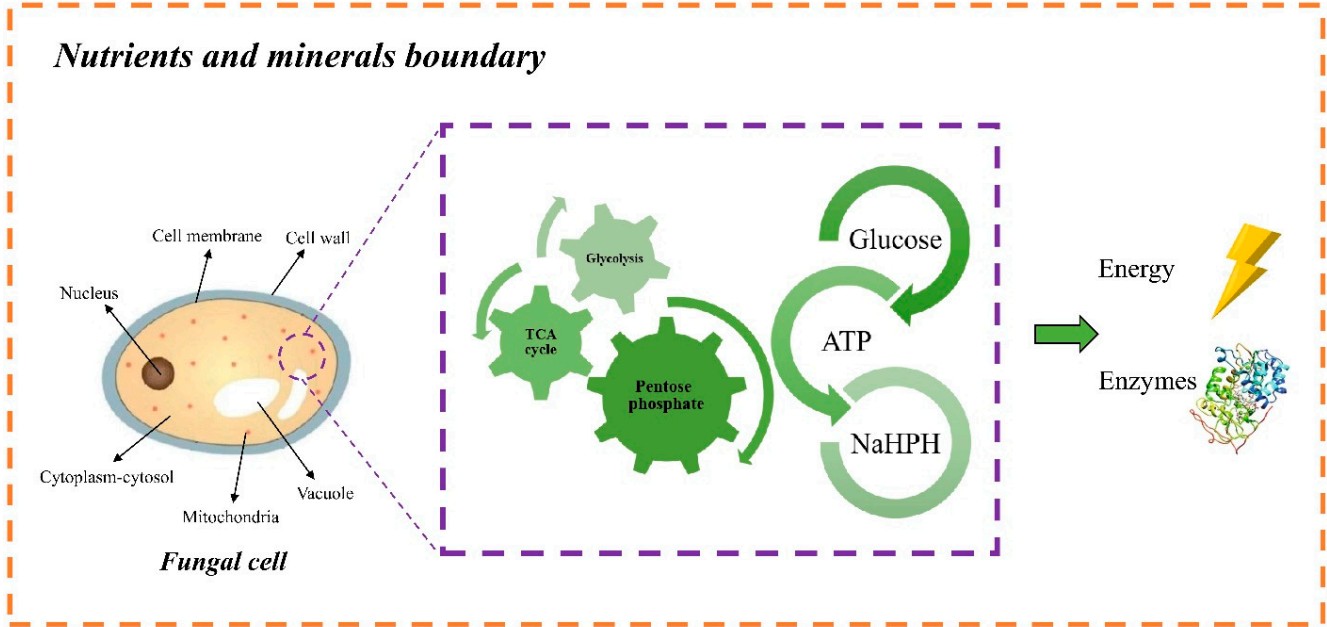

**Figure 7.** Simplified representation of part of the metabolic pathways in cells of white-rot fungi.

Enzyme activity for both LiP and MnP was detected in all the samples. Nonetheless, the enzyme concentration was lower when malt extract was replaced by corn steep at concentrations ≥1.75 g/L (2 U/L LiP and 11 U/L MnP), and yeast extract was substituted for corn steep at concentration ≥3.75 g/L (9 U/L LiP and 29 U/L). The maximum enzyme concentration (33 U/L LiP and 98 U/L MnP) was produced when yeast extract was substituted by corn steep in the CM in the range of 1.25 to 2.5 g/L, reaching a sulfide oxidation of 18.2% after 14 d bio-oxidation. The general trends can be interpreted by analyzing the general composition of each particular component in the CM. About 90% of malt extract is composed of reduced sugars, including but not limited to maltose, fructose, and sucrose, with a lower proportion of nitrogenous substrate, e.g., amino acids and peptides. On the other hand, yeast extract is a good source of N and other essential micronutrients for the better functioning of fungal cells comprising vitamins (biotin and vitamin B), TE, as well as proteins, free amino acids, and peptides. The nourishing composition of yeast extract makes it a valuable medium ingredient.

Based on the composition of corn steep, this substrate is also considered a source of N with nearly 7.7–8.2% of total N [44,45]. Similarly to yeast extract, corn steep is also a source of TE and vitamins. Additionally, it has a significant content of lactic acid, representing 10–30% of the dry mass and sugars, including glucose and fructose (below 5%) [41]. Hence, corn steep is a substrate with a composition similar to yeast extract, providing N sources, vitamins, and TE required to sustain the fungal growth and metabolism, resulting in comparable outcomes when not supplying yeast extract in the CM. Consequently, corn

steep could be considered to replace yeast extract; however, it resulted in an unsatisfactory response as a replacement of malt extract.

### 3.6. CM Comparison

The modified concentrations of reagents to attain maximal sulfide oxidation found in prior sections (12.86 g/L glucose, 2.20 g/L malt extract, 1.67 g/L yeast extract, and 0.49 g/L $MgSO_4 \cdot 7H_2O$) were utilized as a reference for the substitution of yeast extract for corn steep in subsequent bio-oxidation experiments. In the modified CM containing yeast extract, enzyme activity was 58 U/L LiP and 143 U/L MnP after 7 d of growth, and sulfide oxidation after 14 d bio-oxidation was 28.7%. When replacing yeast extract for corn steep in the modified CM, enzyme activity resulted in 39 U/L LiP and 116 U/L MnP, and sulfide oxidation was 21.6% following 14 d of bio-oxidation, as shown in Figure 8.

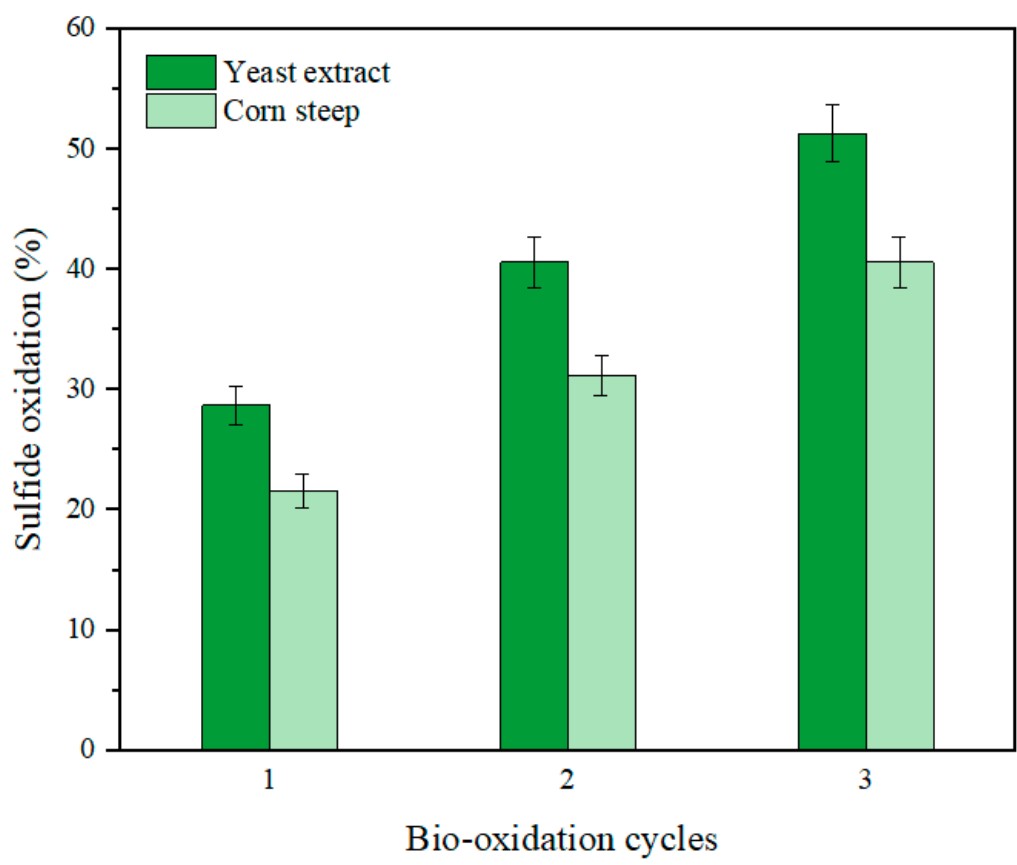

**Figure 8.** Sulfide oxidation was obtained after (1) first cycle of bio-oxidation, (2) second cycle of bio-oxidation, and (3) third cycle of bio-oxidation with *P. chrysosporium* using modified CM with yeast extract and corn steep at 5% *w/v* PD and 37 °C.

Since sulfide oxidation stalled after 14 d of bio-oxidation with no considerable increase after extended reaction time, a replenishing strategy was used to further compare both CM in terms of sulfide oxidation as well as Au recovery [6]. A comparison between the modified CM having equal concentration of yeast extract and corn steep is displayed in Figure 8., showing sulfide oxidation after three cycles of 14-d bio-oxidation. Results revealed that sulfide oxidation gradually increased with the number of replenishing cycles up to 51.1% and 40.6% in the modified CM containing yeast extract and corn steep, respectively. Table 4 shows the Au recovery obtained from the oxidized ore after three cycles of bio-oxidation for a total of 42 d, resulting in 52.1% and 43.8% using the modified CM having yeast extract and corn steep, correspondingly.

**Table 4.** Comparison between yeast extract and corn steep, including composition, cost, and Au recovery from the oxidized ore after replenishing bio-oxidation for three cycles [44,45].

| Factor | Yeast Extract | Corn Steep |
|---|---|---|
| Total nitrogen (%) | 10–11.8 | 7.7–8.2 |
| Average cost of reagent (USD/ton) | 6000 | 375 |
| Au recovery after 3 cycles of bio-oxidation (%) | $52.1 \pm 2.7$ | $43.8 \pm 2.4$ |

Comparing the results using yeast extract and corn steep in the modified CM, the former led to a slightly higher Au recovery of ca. 52.1%, whilst the latter could represent a more economical alternative. Corn steep is a waste product originating from corn industry plants and thus could be considered as a low-cost replacement ingredient for the CM of *P. chrysosporium*. When comparing the cost of reagent in bulk, yeast extract turns out to be 16 times more expensive on average, considering the same mass of corn steep, as shown in Table 4.

Due to the similar results in sulfide oxidation and Au recovery obtained in both CMs, it might be worth exchanging a slightly higher Au recovery for a more economical operational cost, considering low-budget reagents in replenishing bio-oxidation. Still, it is worth noting that sulfide oxidation <30% after one cycle of bio-oxidation is not significantly comparable to the conventional bio-oxidation process with acidophilic bacteria. Nevertheless, it should be noted that this research, for the first time, investigates the use of corn steep in bio-oxidation of low-grade refractory Au ores, and therefore, the process needs further research. There could still be other additional factors for further development to improve the process, such as replacing other components for waste products, e.g., glucose for molasses, and utilizing chemostat-like reactors with multiple sensors to keep an optimum level for nutrients, pH as well as metabolites, which could maximize sulfide oxidation and Au recovery in future research.

### 3.7. Implications and Limitations

This study shows a dependency of the fungus *P. chrysosporium* on CM conditions, including mainly the nutrient availability and pH solution >5.8, leading to particular microbial activities and, subsequently, different trends in sulfide oxidation. Consequently, this research highlights that both a limitation and an excess of nutrients can restrict the performance of this microorganism and, therefore, sulfide oxidation. An adequate concentration of reagents is essential in bio-oxidation experiences, especially glucose, malt extract, and yeast extract, which were more relevant for the fungal activity. Another important implication of this investigation was the potential of corn steep waste to support microbial growth by serving as a source of micro- and macro-nutrients. This implication allows evaluating other waste or less refined products, e.g., molasses, to find a more cost-effective option to oxidize sulfidic ores. From an economic perspective, a full operation using waste products as a source of nutrients to grow the microorganism could considerably decrease the cost associated with the CM, increasing the viability of the process compared to the standard CM option. The use of waste containing CM would further strengthen the circularity of waste management, contributing to a decrease in the accumulation of municipal solid waste and global greenhouse emissions worldwide [23].

The current investigation had certain limitations. Although corn steep waste supported the microbial activity of the fungus *P. chrysosporium* to oxidize a sulfidic Au ore, the pH of the solution remained acidic throughout the tests. Likely, the composition of corn steep based on 10 to 30% *w/w* lactic acid could be the principal responsible for that pH trend compared to the yeast extract-bearing CM. In addition, the performance of the fungi still needs to be further strengthened to improve the bio-oxidation of sulfide ores compared with the conventional option assisted by acidophilic bacteria. The concentration of metabolites is a limitation that might also have restricted further oxidation. Therefore, more research should be done in that direction by considering bioengineered microorganisms that would improve the fungal activity under the studied experimental conditions [4].

The increase in the concentration of metabolites at different pH ranges using current and new reagents, especially waste products, considering chemostat-like reactors, should also be evaluated to overcome these limitations.

## 4. Conclusions and Future Perspectives

Bio-oxidation of a sulfidic Au ore assisted by *P. chrysosporium* was conducted to elucidate the effect of each CM component and subsequently maximize sulfide oxidation using D-optimal RSM. The concentration of glucose, yeast extract, and malt extract were more relevant in the range 10–15 g/L, 1.25–2.5 g/L, and 1.75–3.5 g/L correspondingly and favored microbial activity with $\geq$51 U/L LiP and $\geq$128 U/L MnP and sulfide oxidation $\geq$23%. There was a negligible influence of the $MgSO_4 \cdot 7H_2O$ concentration in the range of 0.0–1.0 g/L. On the basis of the model suggested by D-optimal RSM, 12.86 g/L glucose, 2.20 g/L malt extract, 1.67 g/L yeast extract, and 0.49 g/L $MgSO_4 \cdot 7H_2O$ was proposed as modified CM to produce almost 29% sulfide oxidation after 14 d of bio-oxidation. Corn steep waste supported microbial activity when replacing yeast extract in the CM. The modified CM at which yeast extract was replaced with 1.67 g/L corn steep led to 40.6% sulfide oxidation after the three-cycle bio-oxidation for 42 d with a 43.8% Au recovery, comparable with the CM containing yeast extract. Despite this study showing that corn steep waste has the potential to sustain the microbial activity of *P. chrysosporium* to oxidize sulfidic ores, it is still necessary to further investigate the use of other waste products following more development to increase enzyme concentration in desired pH environments. It is also recommended assessing this bio-based system with tools related to life cycle assessment, energy, and economics, which can further help to strengthen the process considering the life cycle of products and residues.

**Supplementary Materials:** The following supporting information can be downloaded at: https://www.mdpi.com/article/10.3390/su152115559/s1. Figure S1: (a) particle size distribution and (b) XRD pattern of the gold-bearing sulfidic ore used in bio-oxidation experiments. Table S1: D-optimal RSM test matrix for parameter optimization for bio-oxidation assisted by *P. chrysosporium.* Table S2. ANOVA results for the fitted multiple regression model.

**Author Contributions:** Formal analysis, G.H.; Investigation, G.H.; Data curation, G.H.; Writing—original draft, G.H.; Writing—review & editing, H.M. and A.G.; Supervision, H.M and A.G.; Project administration, A.G.; Funding acquisition, A.G. All authors have read and agreed to the published version of the manuscript.

**Funding:** This work was supported by COREM and MITACS through a collaborative research grant MITACS Accelerate Fund IT13524.

**Institutional Review Board Statement:** Not applicable.

**Informed Consent Statement:** Not applicable.

**Data Availability Statement:** No new data were created or analyzed in this study. Data sharing is not applicable to this article.

**Conflicts of Interest:** The authors declare that they have no known competing financial interests or personal relationships that could have appeared to influence the work reported in this paper.

## Abbreviations

| | |
|---|---|
| ANOVA | Analysis of variance |
| ATCC | American-type culture collection |
| ATP | Adenosine triphosphate |
| Au | Gold |
| Bio-oxidation | Biological oxidation |
| C | Carbon source |
| CM | Culture medium |
| DOE | Design of experiments |

| Sf | Final sulfur content |
| Si | Initial sulfur content |
| LiP | Lignin peroxidase |
| MnP | Manganese peroxidase |
| NADPH | Nicotinamide adenine dinucleonide |
| N | Nitrogen source |
| ND | not detected |
| NR | not reported |
| OD | Optical density |
| ORP | Oxidation-reduction potential |
| *P. chrysosporium* | *Phanerochaete chrysosporium* |
| *p*-value | probability value |
| PD | pulp density |
| RSM | Response surface methodology |
| TE | Trace elements |
| TCA | Tricarboxylic acid |

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
