# Peer review of "Assessment of Modified Culture Conditions for Fungal Bio-Oxidation of Sulfidic Gold Ores Performed at Circumneutral pH"

_sustainability, doi:10.3390/su152115559_

Round 1
Reviewer 1 Report
Comments and Suggestions for Authors
Sustainability
Manuscript ID: sustainability-2666048
Assessment of culture medium and utilization of corn steep waste in fungal treatment of gold bearing sulfidic ores at circumneutral pH
The work could be of general interest. The following comments should help further improve the quality of the work:
1-The title is not suitable; it is a bit lengthy and does not read well either.
2-Latin words such as “in-situ” should be italicized.
3-Please avoid abbreviating terms in the Abstract, especially those that have been used once only.
4-Throughout the main body of the manuscript, please avoid abbreviating terms which are used once or twice. Only abbreviate terms if they are used three times or more.
5-Please add a table of abbreviations.
6-Please add two more keywords (generally, up to 6 is allowed). Metadata, including keywords, are important in terms of the searchability of the manuscript if published.
7-The novelty/originality of the paper should be more effectively established. It would be advisable to add a Table to the “Introduction” section, tabulating the latest works in the field to highlight the novelty of the present work accordingly.
8-Circularity of waste management is an important topic and the present work is in alignment with that, and should be further highlighted and explained. Some recent reports have elaborated on the significance of bio-based solutions to increase the circularity of waste management. Here is an example “Boosting the circularity of waste management: pretreated mature landfill leachate enhances the anaerobic digestion of market waste”, which if found suitable by the authors, can be considered.
9-The quality of Figure 1 is low; it is blurred in its present form. Please improve it.
10-Please make sure all the units will be presented in compliance with the SI System. For instance, please use "d" for "days", etc. This comment applies to Figures/Tables, too.
11-The quality of Figure 2 is low; it is blurred in its present form. Please improve it.
12-Reference lumping should be avoided. Please cite references where they exactly belong; this will prevent reference lumping.
13-The quality of Figure 3 is low; it is blurred in its present form. Please improve it.
14-The same goes for Figures 4, 5, and 6. The quality of Figures is generally low; they are blurred in their present forms. Please improve them for both resolution and graphical features.
15-Bioproduct-based systems such as the one presented herein should also be investigated for their sustainability. Hence, future studies should further investigate the findings presented herein from the sustainability perspective using advanced sustainability assessment tools, including life cycle assessment etc., as elaborated in recent works such as “Life cycle assessment for sustainability assessment of biofuels and bioproducts”, etc. Authors can briefly discuss this future research need using works such as the example provided, but not necessarily limited to that, and highlight the importance of such additional assessments to direct future studies.
16-Please change “4. Conclusion” to "4. Conclusions and future perspectives". Accordingly, please also elaborate on the future research needs in this domain. Moreover, please include the main numerical results in this section.
17-More comparison of the results obtained with those of previous studies and critical discussion should also be added.
18-Please include and discuss the limitations of the present study as well.
19-The practical implications of the present study should be included as well.
Reviewer 2 Report
Comments and Suggestions for Authors
My comments are detailed below:
The abstract should be more informative.
The English writing is good and some corrections are needed.
Line 9: Please change “originated” to “originating”
Line 32: Please change “to deal” to “in dealing”
Line 49: Please change “fungus” to “the fungus”
Line 96: Please change “as principal” to “as the principal”
Line 132: Please change “prior to” to “before”
Line 158: Please change “prior commencing” to “prior to commencing”
Line 202: Please change “prior to” to “before”
Line 566: Please change “of” to “for”
Line 587: Please change “has” to “have”

Comments on the Quality of English LanguageThe abstract should be more informative.
The English writing is good and some corrections are needed.
Reviewer 3 Report
Comments and Suggestions for Authors
This study is worthy of investigation having some practical implications on the recovery of valuable resources like gold from the gold-containing sulfidic ores. However, the manuscript needs considerable revision to improve the overall quality prior consideration for publication. The additional comments are presented below.
Comments:
Abstract:
Line 12: “a white-rot fungus…” Also, mention the name of the fungus.
Line 92: This study was done using a white-rot fungus “P. chrysosporium”. Could you explain why you have selected this specific fungal strain, why not Aspergillus niger which is widely used in waste treatment and resource recovery.
A few studies have reported use of bacterial leaching for treatment of gold-containing sulfidic ores. What are the potential advantages of using fungus-based method compared to bacterial-based method for treatment of gold-containing ores and recovery of gold?
In the introduction, the justification on the novelty and importance of this study with respect to the existing knowledge in literature can be strengthened.
Line 109: “A sulfidic gold ore classified…” State the sources of the ore used in this work.
Table 2: Why the unit for concentration of different elements is not uniform (ppm vs %). It would be better to be in a uniform unit.
Authors are suggested to use some statistical analysis to show whether the difference in the gold recovery using different substrates (e.g., Table 2) is statistically significant.
For better understanding to the readers, the entire processes including the treatment and gold recovery can be presented in a schematic diagram.
Figure 6: Usually fungal metabolism using organic substrates produces diffident metabolites, specifically different carboxylic acids. Any information about the dominant carboxylic acid produced using the yeast extract or corn steep as the substrates here.
The discussion on the potential mechanisms for the bio-oxidation of sulphide-containing ores needs to be strengthened.
At the end of results and discussion, add a section on the implications and limitations of this work.
The conclusion is too long and not focused, thus it can be condensed by highlighting the important findings and observations from this study.
Table 3: I think, the recovered gold is still in the liquid solution? If yes, what ate potential methods to transfer into solid form.
From the practical application point of view, what would be the quality (e.g., purity) of the recovered gold from the fungal treatment?
Delete the pronouns like “we” throughout the manuscript and change the text accordingly.
Comments on the Quality of English LanguageMinor English editing.
Reviewer 4 Report
Comments and Suggestions for Authors
Dear authors, I have carefully studied your article and I have several suggestions for improving it.
1. The purpose of this study should be more clearly stated in the Abstract and Introduction.
2. At the end of the Introduction section, it is necessary to specify the further structure of the article.
3. Removing links before 2000 and adding more links for 2022-2023 is advisable. The total number of links may be reduced.
4. The stages of biooxidative experiments should be presented in the form of a flow chart.
5. It is necessary to divide the Results and Discussion section into 2 different sections.
6. In section 2, a subsection on statistical analysis is necessary.
7. The format of the drawings must be unified.
Round 2
Reviewer 1 Report
Comments and Suggestions for Authors
The manuscript can be accepted for publication.
Reviewer 3 Report
Comments and Suggestions for Authors
The quality and readability of the revised manuscript is improved after revision. Thus, it can be considered for publication.
Comments on the Quality of English LanguageMinor English Editing.